# Current induced electromechanical strain in thin antipolar Ag$_2$Se semiconductor

Hao Luo[1,2,7], Qi Liang[1,2,7], Anan Guo[1,2], Yimeng Yu[1,2], Haoyang Peng[1,2], Xiaoyi Gao[1], Yihao Hu[3], Xianli Su[1], Ctirad Uher[4], Yu Zheng[1], Dongwang Yang[1], Xiaolin Wang[5], Qingjie Zhang[1], Xinfeng Tang[1], Shi Liu[3], Gustaaf Van Tendeloo[2,6], Shujun Zhang[5] ✉ & Jinsong Wu[1,2] ✉

Electromechanical coupling permits energy conversion between electrical and elastic forms, with wide applications[1,2]. This conversion is usually observed in dielectric materials as piezoelectricity and electrostriction[3–7]. Electromechanical coupling response has also been observed in semiconductors[8], however, the mechanism in semiconductors with a small bandgap remains contentious. Here we present a breakthrough discovery of a giant electromechanical strain triggered by the electric current in thin antipolar Ag$_2$Se semiconductor. This phenomenon is made possible by the alteration of dipoles at a low current density (step I), followed by a phase transition under a moderate current density (step II), leading to a local strain of 6.7% measured by in-situ transmission electron microscopy. Our finding demonstrates that electric current has both thermal and athermal effect (*e.g.* alteration of dipoles and interaction of dipole vortices with the electric current). This strain allows for the concurrent control of electroelastic deformation and electric conductivity.

Electromechanical coupling is primarily manifested as a piezoelectric effect in non-centrosymmetric dielectric materials, which combines polarization and strain to enable the conversion between electrical and elastic energy[7]. While the electrostrictive effect is present in all-dielectric materials[9–11], the resulting electrostrictive strain is typically insufficient to be used, with the exception of relaxor ferroelectrics which have a giant dielectric permittivity. The electromechanical coupling is also found in ionic electroactive polymers and elastomers, termed ionic actuators, in which the movement of negative and positive ions under an applied electric field generates mechanical deformations[12,13]. Although the electromechanical effect has been recognized in semiconductors for some time[14–16], the underlying mechanism remains unclear, with only a few speculative explanations.

One such hypothesis, 'current striction', suggests that a field-induced shift in the electron distribution in reciprocal space increases electron energy, leading to lattice deformation in large bandgap semiconductors like Ge[17]. For semiconductors with a medium bandgap -2 eV, such as lead halide perovskite, lattice deformation has been attributed to the generation of defects under an applied bias[18]. Piezoelectricity has also been observed in semiconductors, especially in heterostructures, due to a built-in electric field in the charge depletion regions[19]. In contrast, for low bandgap semiconductors, where a large electric current can be created by a small voltage, it remains uncertain whether electromechanical coupling can occur. Nevertheless, the application of the electromechanical effect in nanoscale actuators requires the generation of a substantial elastic strain at a moderate

[1]State Key Laboratory of Advanced Technology for Materials Synthesis and Processing, Wuhan University of Technology, Wuhan, China. [2]Nanostructure Research Center, Wuhan University of Technology, Wuhan, China. [3]Key Laboratory for Quantum Materials of Zhejiang Province, Department of Physics, School of Science and Research Center for Industries of the Future, Westlake University, Hangzhou, Zhejiang, China. [4]Department of Physics, University of Michigan, Ann Arbor, MI, USA. [5]Institute for Superconducting and Electronic Materials, Faculty of Engineering and Information Sciences, University of Wollongong, Wollongong, Australia. [6]EMAT (Electron Microscopy for Materials Science), University of Antwerp, Antwerp, Belgium. [7]These authors contributed equally: Hao Luo, Qi Liang. ✉e-mail: shujun@uow.edu.au; wujs@whut.edu.cn

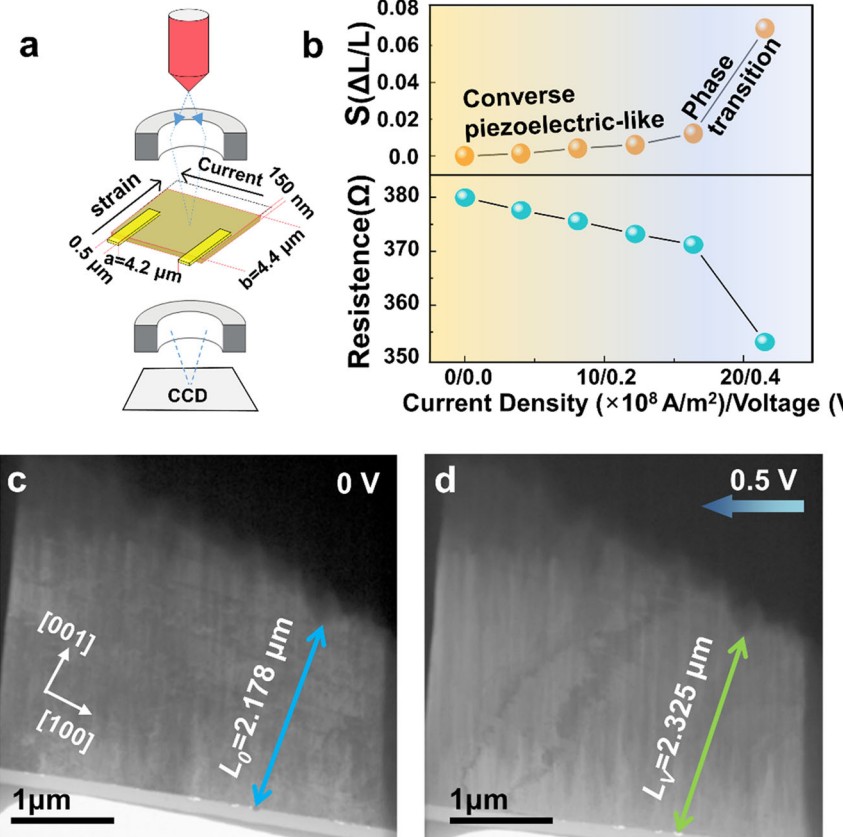

**Fig. 1 | A giant electromechanical coupling effect is observed in Ag$_2$Se film by in-situ TEM. a** Illustration of the experimental setup in measuring the electric current and electromechanical strain conducted by an in-plane capacitor geometry. The strain is measured by directly recording the TEM images of the Ag$_2$Se slice under the applied voltages. **b** Strain of the sample measured along the [001] direction and the resistance versus the measured current density and the applied voltage. **c** TEM image of the [010] oriented $\alpha$-Ag$_2$Se at 0 V, where $L_O$ is measured. **d** TEM image of the same region when the applied voltage is 0.5 V, where $L_v$ is measured.

electric field[20], while under certain scenarios electromechanical actuators need to possess desirable electric conductivity. Therefore, there is a strong need to explore alternative electromechanical coupling mechanisms that can meet these requirements in low-band-gap semiconductors.

$\alpha$-Ag$_2$Se is classified as an *n*-type semiconductor characterized by a narrow bandgap of <0.2 eV, which has been intensively studied as a thermoelectric material[21–25], and holds great promises for applications in wearable and flexible thermoelectric devices[26,27]. At a temperature of ~400 K, the orthorhombic $\alpha$-Ag$_2$Se transforms into the cubic $\beta$-Ag$_2$Se phase, exhibiting superionic Ag-conductivity. Interestingly, this temperature-induced phase transition is accompanied by a large elastic deformation[28]. The $\alpha$- to $\beta$-Ag$_2$Se phase transition involves a rearrangement of the Ag$^+$-ions while the Se$^{2-}$-sublattice only slightly alters, which can also be characterized as a topotactic transition[29]. Given the seamless integration of nanoscale Ag$_2$Se thin films into contemporary microelectronic chips, the ability to electrically trigger elastic deformation could significantly enhance its adaptability across a wide range of applications. The narrow bandgap of Ag$_2$Se, which often characterizes it as a semimetal, leads to a large electric current upon application of a small voltage. This makes the exploration of the correlation between electric current and elastic lattice deformation in low bandgap semiconductors particularly intriguing.

In this study, we present compelling evidence of a giant electro-mechanical strain generated by electric current in the Ag$_2$Se thin crystals, which is attributed to the current-induced reorganization of the dipoles and a phase transition. Remarkably, our findings reveal that an electric current of ~1.42 mA (with a current density of approximately $2.15 \times 10^9$ A/m$^2$ and an electric field of ~$1.2 \times 10^5$ V/m) can induce a strain

of ~6.7%. We demonstrate that the $\alpha$-Ag$_2$Se thin crystal possesses spontaneous, antiparallel polarization at room temperature, resembling antiferroelectricity. In addition, we observe that the current not only induces a thermal effect by raising the temperature but also produces an athermal effect, namely the reshaping of the dipole, which is similar to the effect of applying an external electric field. In addition to this newly revealed giant electric current-induced strain, Ag$_2$Se maintains its electrical conductivity throughout the entire study process, along with its inherent flexibility, highlighting its great potential in flexible electroactive devices.

## Results
### Giant electromechanical strain observation in Ag$_2$Se
We synthesized Ag$_2$Se crystals using a Se-vapor transfer method and cut the crystals into thin slices with an averaged thickness of about 150 nm (to ensure a smooth contextual transition, we'll use Ag$_2$Se thin film in the following), via focused-ion beam (FIB) processing. Electrical (current versus applied voltage *V*) and electromechanical (strain *s* versus electric current density *J*) measurements on the Ag$_2$Se films were conducted using an in-plane capacitor geometry, as shown in Fig. 1a. We measured the strain by directly recording transmission electron microscopy (TEM) images of the Ag$_2$Se thin film under the applied voltages, represented by $s = (L_V – L_O)/L_O$, where the measured length changes from $L_O$ to $L_V$ when the applied voltage increases from 0 to 0.5 V (corresponding to a current density of ~$2.15 \times 10^9$ A/m$^2$, Fig. 1c, d). Large electromechanical displacements in the Ag$_2$Se film were experimentally observed. As shown in Fig. 1b, Fig. S1, and Supplementary Movies 1, we observed a linear relationship between the measured strain and the electric current density *J* ($J = \sigma E$, $E = V/d$, where

$d$ is the distance between the two electrodes and $\sigma$ is the conductivity of Ag$_2$Se), then progressively increasing the voltage from 0 V to 0.4 V in steps of 0.1 V. In the subsequent sections, we will use voltage as the driving parameter, as we directly adjusted the applied voltage in the experiment rather than the current density. Considering the temperature rise due to Joule heating, the strain can be expressed using the formula $s = kJ + \alpha\Delta T$ ($k$ is defined as the current-to-strain conversion coefficient, $\alpha$ represents the thermal expansion coefficient and $\Delta T$ is the change of temperature). The temperature change $\Delta T$ typically exhibits a nonlinear relationship with the applied voltage. Given the linear relationship observed between the measured strain $s$ and both the electric current density $J$ (Fig. 1b) and the applied voltage V, i.e., $s \sim k\sigma E = k\sigma V/d$, it becomes evident that the current density $J$ exerts an athermal effect on the strain. This can be substantiated by the fact that the temperature variations (the thermal effect of electric current) typically exhibit a quadratic relationship with current density in low bandgap semiconductors. The athermal effect, which is comparable to a piezoelectric effect, predominantly governs the observed strain in Ag$_2$Se thin film below 0.4 V.

A particularly remarkable finding is the attainment of a giant strain, ~6.7%, at a voltage of 0.5 V (corresponding to a current of ~1.42 mA and a current density of ~$2.15 \times 10^9$ A/m$^2$). Using in-situ TEM, we can identify the microstructural evolution (Fig. S1) and phase transition in the Ag$_2$Se thin films (Fig. S2), along with the measurement of electromechanical strain. Selected-area electron diffraction (SAED) reveals that the Ag$_2$Se film is oriented along the [010] zone axis, and the largest electromechanical strain occurs along the [001] direction. Our findings further reveal a notable phase transition from $\alpha$-Ag$_2$Se [010] to $\beta$-Ag$_2$Se [011] occurs at an applied voltage of 0.5 V (Fig. S2), leading to the generation of a substantial electromechanical strain. Consequently, the observed elastic deformation in Ag$_2$Se can be categorized into two distinct steps: an initial deformation originating from the athermal effect of electric current akin to piezoelectricity and a subsequent deformation resulting from the electric current-induced phase transition.

To exclude the thermal influence from electron beam heating, the experiments were repeated at low beam intensity and low temperature using a cryo- and biasing in-situ TEM holder, which allowed for the clear observation of both the athermal and thermal effects of the current. As illustrated in Fig. S3 and Supplementary Movies 2, the sample temperature remained below 233 K even at a high current density of ~$1.09 \times 10^{10}$ A/m$^2$ (when the $\alpha$ to $\beta$ phase transition occurs), due to the continuous cooling by liquid N$_2$ during TEM observation. This temperature is much lower than that required for temperature-induced phase transition, i.e., ~407K[23]. This highlights the phase transition can be triggered either by temperature (thermally induced phase transition), or the electric current alone (current athermal effect inducing phase transition).

At room temperature, the temperature rise of Ag$_2$Se sample during the current application was monitored by high-precision infrared thermometry equipment. As depicted in Fig. S4, the sample temperature reaches 351 K when $\alpha$-Ag$_2$Se completely transforms into $\beta$-Ag$_2$Se at $V = 0.55$ V. Despite evident Joule heating effect (with the sample temperature increasing by 53 K) at higher current density, it remains far below the threshold required for temperature-induced phase transition. This showcases both the thermal and athermal effects of the electric current, with the latter playing the dominant role in this phase transition. It is noteworthy that under increased current density, the crystal undergoes an elongation in morphology. Simultaneously, there are modest variations in electric resistance, marked by a decrease from 380 to 350 $\Omega$ at the voltage corresponding to the phase transition (Fig. 1b). To further explore the interaction between electronic conductivity and ionic conductivity in Ag$_2$Se, we tested the dielectric properties, carrier mobility, and carrier concentration, as shown in Figure S5. As seen in Fig. S5a, when the temperature reaches

~360 K, the free carrier concentration of Ag$_2$Se increases sharply, accompanied by a decrease in carrier mobility. This phenomenon is attributed to the migration of Ag$^+$ ions, which scatter electrons and thus affect electrical conductivity. From 298 K to 373 K, the Nyquist plot shows a single semicircle, indicating that electronic conduction dominates. Above 373 K, the high-frequency region of the Nyquist plot remains semicircular, while a diffusion tail appears in the low-frequency region (Warburg impedance), suggesting the onset of ionic migration. Thus, below 373 K, the main conduction mechanism is electronic, while above 373 K, it becomes a hybrid of electron and ion conduction (Fig. S5b).

The phase transition and the associated electromechanical strain are largely reversible when reducing the applied voltage, as depicted in Fig. S6 and Supplementary Movies 3. Conversely, by reversing the polarity of the applied voltage, we replicated an identical phase transition from $\alpha$- to $\beta$-Ag$_2$Se along with the induced electromechanical strain, as given in Supplementary Movies 4. Of particular significance is that we observed a swift and reversible phase transition and the associated strain in the Ag$_2$Se films (Fig. S7 and Supplementary Movies 5) under an alternating current (of ~0.1–0.2 Hz). The phase transition between the $\alpha$ and $\beta$ phase was accompanied by alterations in the measured resistance (Fig. S8), morphological variation (Supplementary Movies 6), and structural evolution (Supplementary Movies 7 and Fig. S8a). Limited by the temporal resolution of the CCD camera, images were collected every 50 ms. We thus observed the electromechanical strain at a frequency of 20 Hz (Supplementary Movies 8). However, the phase transition and its related resistance alternation can be monitored at a higher frequency, such as 5000 Hz, as shown in Fig. S9, where the resistance variations (which is a sign of phase transition) match the frequency of the alternative voltage. It is important to emphasize that the instantaneous response of the strain and phase transition to the electrical pulse underscores the athermal effect of the electric current. In contrast, slower thermal effect, such as Joule heating within the sample, has a smaller impact on the observed phenomena[30].

The strain was further verified by macroscopic tests (Fig. S10), which showed a strain of about 1.8%. In these tests, a voltage of 10 V was applied to the sample, producing observable strain and elastic deformation at a temperature of 346.4 K, which is significantly lower than the theoretical phase transition temperature of 407 K. Additionally, piezoelectric force microscopy (PFM) experiments were carried out. Figure S11 presents the amplitude and phase signal, confirming the presence of strain under the applied electric field.

## The existence of antiparallel spontaneous dipoles in Ag$_2$Se

To gain insight into the underlying mechanism responsible for the observed large electromechanical coupling effect and the reversible phase transition triggered by electric current, we conducted a comprehensive study of the microstructure using the scanning TEM (STEM) technique. Our exploration reveals the presence of spontaneous and antiparallel polarization in the structure of $\alpha$-Ag$_2$Se. In the high angle annular dark field (HAADF) STEM images of $\alpha$-Ag$_2$Se (orthorhombic, $a = 4.3357$ Å, $b = 7.07$ Å, $c = 7.774$ Å, space group of $P2_12_12_1$)[31], the position of each Ag$^+$ and Se$^{2-}$ column can be accurately identified along the [100] zone axis. Analysis of these images unveils that the Se$^{2-}$ position does not coincide with the center of the Ag$^+$ tetrahedron, as illustrated in Fig. 2a, where the center of the Ag$^+$ tetrahedron is represented by a gray dot, while the Se$^{2-}$ position is indicated by a blue dot. This observation demonstrates the existence of a spontaneous polarization in $\alpha$-Ag$_2$Se at room temperature. The polar vectors of the dipoles are [01$\bar{4}$]/[0$\bar{1}$4] and [014]/[0$\bar{1}\bar{4}$] (as shown by the blue arrowheads in Fig. 2b). Consequently, pairs of oppositely oriented polarization form antiparallel dipoles in the crystal, as illustrated in Fig. 2c. First-principles density functional theory (DFT) calculations confirm that the displacement of Se anions primarily occurs along the

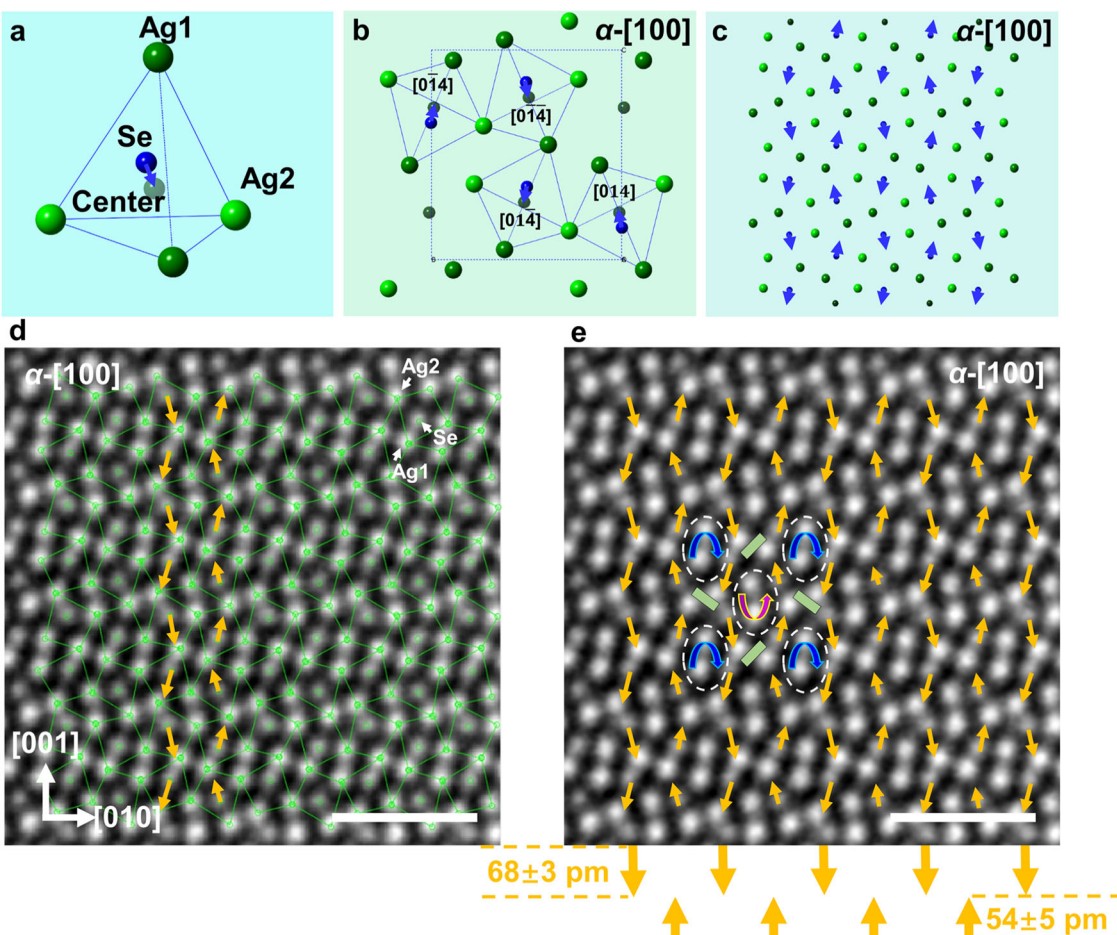

**Fig. 2 | Spontaneous and antiparallel polarization in $\alpha$-Ag$_2$Se. a** In the Se-Ag tetrahedron, the basic structural unit of $\alpha$-Ag$_2$Se, the center of Ag$^+$-cations deviates from the Se$^{2-}$ anion position, inducing a localized polarization. **b** Projection of $\alpha$-Ag$_2$Se along the [100] zone axis showing the arrangement of the polarizations in a unit cell. **c** Pairs of oppositely oriented polarizations with vectors of [01$\bar{4}$]/[0$\bar{1}$4] and [014]/[0$\bar{1}\bar{4}$] are formed, like a regular arrangement of antiparallel dipoles. **d** HAADF STEM image of the [100] oriented $\alpha$-Ag$_2$Se, in which the Ag$^+$-tetrahedra are outlined and the localized and antiparallel dipoles can be experimentally measured, as indicated by the arrowheads. **e** Analysis of the map of polarizations shows that the displacement vectors pointing upward (close to the [001] direction) have an average length of ~54 pm, which is shorter than those pointing downward (close to the [00-1] direction) with an average length of ~68 pm. Also the alternatively clockwise and anticlockwise vortexes formed by the dipoles can be seen. The scale bar is 1 nm.

<001> direction, simultaneously prompting the neighboring dipoles to adopt the antipolar ordering (Fig. S12). A HAADF STEM image of $\alpha$-Ag$_2$Se along the [100] zone axis is illustrated in Fig. 2d, where each Ag$^+$ and Se$^{2-}$ column can be clearly distinguished, allowing for accurate position determination. By measuring the deviation of the Ag$^+$-ion centers from Se$^{2-}$ (as outlined in the image), the dipoles can be experimentally determined as indicated by the orange arrowheads (Fig. 2d). Furthermore, Fig. 2e demonstrates that the polar vector map can now be interpreted as having alternating clockwise and anticlockwise vortexes, as marked by the blue and purple arrows, respectively. Additionally, convergent dipoles form an interesting topological structure, as depicted by the green bars.

During STEM sample preparation, strain typically accumulates, prompting adjustments in the antiparallel dipoles to accommodate this strain. As observed in Fig. 2e, the averaged magnitude of the dipoles pointing upward (close to the [001] direction) is shorter than the magnitude of those pointing downward (close to the [00$\bar{1}$] direction). Specifically, the average length of the polar vectors near the [001] direction is ~54 ± 5 pm, while those close to the [00$\bar{1}$] direction are ~68 ± 3 pm. The resulting net atomic displacement along the polar direction [00$\bar{1}$] is thus $\Delta z = $ ~14 pm, corresponding to a spontaneous polarization estimated as $P_s = (258 \pm 9) \times \Delta z = 3.6 \pm 0.1\ \mu\mathrm{C\ cm}^{-2}$ [32]. The buildup of a net polarization in the antipolar crystal (which should

have zero polarization in its pristine state) arises from the strain in the Ag$_2$Se crystal, which alters the dipoles in the structure.

## The alteration of polarization under external voltages
We also directly observed the alteration of the antiparallel dipoles in Ag$_2$Se under an external voltage by in-situ STEM, providing crucial insights into the atomic mechanism of electromechanical strain generation under an electric current which is induced by a relatively small voltage (Step I). The experimental setup is illustrated in Fig. S13. HAADF STEM images of $\alpha$-Ag$_2$Se along the [100] zone axis were taken under the applied voltages of 0 V (Fig. 2e, $J = 0$ A/m2) and 0.4 V (Fig. 3a, $J = 1.63 \times 10^9$ A/m$^2$), respectively. The applied voltage, along the [011] direction, consists of two components, one aligned with the [001] direction, and the other along the [010] direction (inset in Fig. 3a). The polar vector map of $\alpha$-Ag$_2$Se at 0.4 V (and $J = 1.63 \times 10^9$ A/m$^2$) is given in Fig. 3a. The polar vector maps at 0 V (orange color) and 0.4 V (purple color) are compared in Fig. 3b, c. Under the current, certain dipoles (with vectors of [0$\bar{1}$4]/[014]) along the current direction exhibit subtle elongation, experiencing an increase in average magnitude from ~54 ± 5 pm to ~58 ± 3 pm. Conversely, those dipoles (with vectors of [01$\bar{4}$]/[0$\bar{1}\bar{4}$]) against the current direction demonstrate a reduction in average magnitude from ~68 ± 3 pm to ~63 ± 6 pm, as shown in Fig. 3c. As the dipoles have a large component along the [001] direction, their

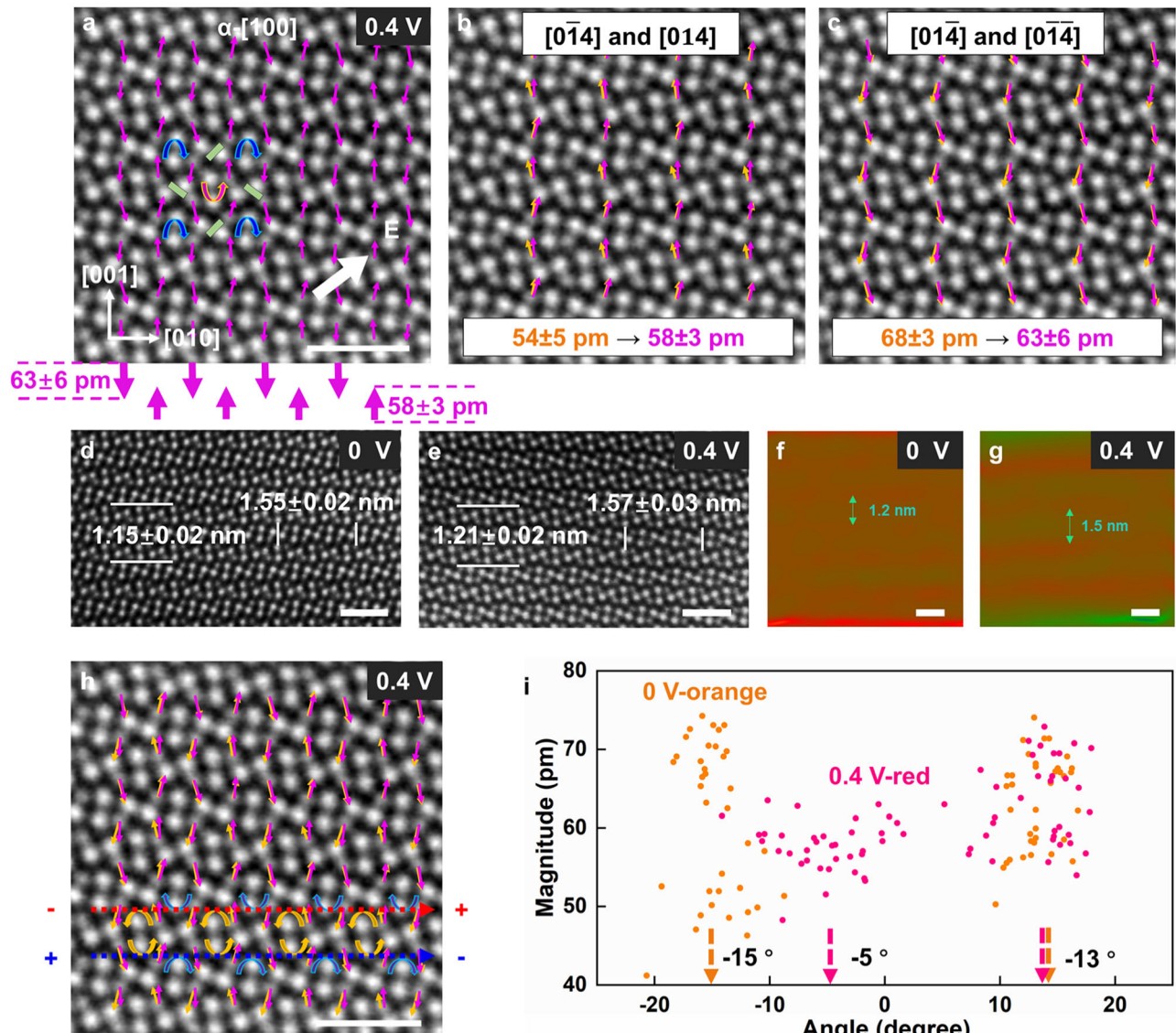

**Fig. 3 | Alteration of the polarizations under the applied current. a** HAADF STEM image of the [100] $\alpha$-Ag$_2$Se and overlapping with the polarizations (in purple) when the applied voltage is 0.4 V. **b** STEM image overlapping with the dipoles oriented along the electric field (those with vectors of [0$\bar{1}$4] and [014]), at 0 V (orange) and 0.4 V (purple). The dipoles are elongated with an average magnitude changing from ~54 pm to ~58 pm. **c** STEM image overlapping with the dipoles oriented against the electric field (those with vectors of [01$\bar{4}$] and/[0$\bar{1}\bar{4}$]), at 0 V (orange) and 0.4 V (purple). The dipoles are shortened (the average magnitude changes from ~68 pm to ~63 pm). **d**, **e** A comparison of the STEM images at 0 V and 0.4 V, showing the expansion of the (010) and (001) lattice plane under the applied current.

**f**, **g** Comparison of the strain map of the (001) plane at 0 V and 0.4 V. **h** STEM image of the [100] $\beta$-Ag$_2$Se overlapping with the dipoles at 0 V (orange) and 0.4 V (purple), in which the localized vortexes are shown. The migration path where the electron's mobility is enhanced is shown by a blue dotted line. The migration path where electron's mobility is weakened is shown by a red dotted line. The dipoles that hinder the current are rotated toward the [010] direction, while those that accelerate the current remain unchanged. **i** A statistical analysis showing the magnitude and angle of the dipoles measured at 0 V and 0.4 V. The dipoles that hinder the current rotate at an average angle of 10° (from ~-15° at 0 V to ~-5° at 0.4 V), while those that accelerate the current do not rotate. All scale bars are 1 nm.

alteration under the electric current along this direction is most pronounced. The (001) lattice spacing increases from 1.15 ± 0.02 nm at 0 V to 1.21 ± 0.02 nm at 0.4 V, revealing a directional elastic stretch (Fig. 3d, e). In addition, when examining the strain map derived from the STEM image using the geometric phase analysis (GPA)[33] method, a nearly periodic strain along the [001] direction is observed in Ag$_2$Se at 0 V, with a period of ~1.2 nm (Fig. 3f). Based on the same analytical procedure, it is observed that the strain pattern undergoes an evolution when subjected to the electric current, with the periodicity changing from ~1.2 nm to ~1.5 nm (Fig. 3f, g), confirming an elastic deformation induced by the electric current. Based on the above observations ($\Delta z$ = ~5 pm), the net polarization becomes $P_E$ = (258 ± 9) × $\Delta z$ = 1.8 ± 0.1 μC/cm$^2$ under the electric current, which is

nearly opposite to the direction of spontaneous polarization $P_s$. This demonstrates that the electric current clearly alters the polarization in the antipolar crystal, resembling a converse piezoelectric effect, highlighting the athermal character of the electric current.

The spontaneous dipoles also possess a small component along the [010] direction, and their alteration under an electric current along this direction is mainly influenced by the configuration of the vortexes. An intriguing observation is that the alternating arrangement of clockwise and anticlockwise vortexes creates two different paths for charge carrier transport. One path, where positive charge carriers are accelerated, is shown by the blue dashed line in Fig. 3h, while the other path, where the positive charge carriers are hindered, is given by the red dashed line in Fig. 3h. Under the applied current, it is found that the

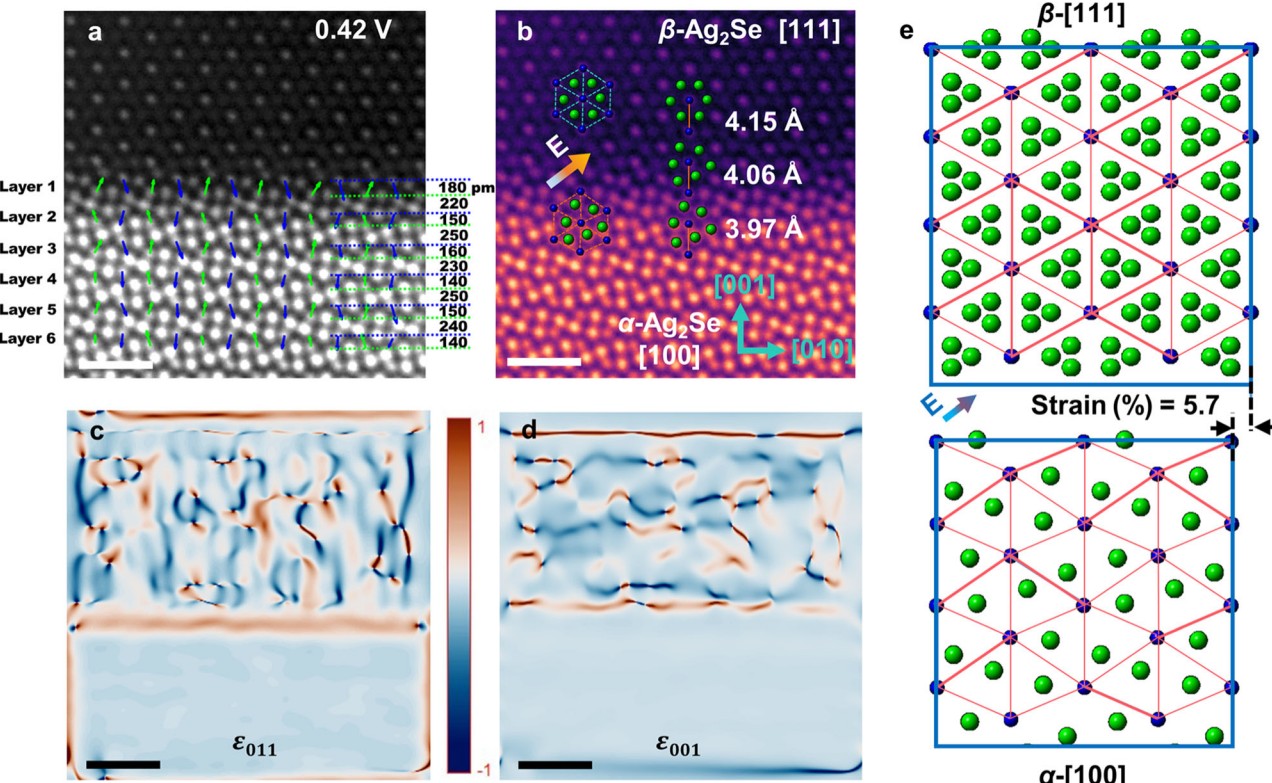

**Fig. 4 | Large polarizations are created at the interface, leading to the $\alpha$-to-$\beta$ phase transition accompanied by a giant elastic expansion of the Se lattice. a** STEM image of the $\alpha$-$\beta$ interface, in which the polarizations and spacing of the (001) Se-lattice of $\alpha$-Ag$_2$Se are labeled. The scale bar is 1 nm. **b** STEM image of the interface, in which the structural units are outlined, showing that the $\alpha$-to-$\beta$ phase transition is a low-symmetry to high-symmetry and topologically polar to non-polar transition. The scale bar is 1 nm. **c, d** Strain maps of the interface showing the strain along the (011) plane and the (001) plane, respectively. The scale bar is 1 nm. **e** Given the [100] $\alpha$-Ag$_2$Se transforming into the [111] $\beta$-Ag$_2$Se, a strain of ~5.7% will be generated along the [001] direction (referring to $\alpha$-Ag$_2$Se).

dipoles hindering the electric current rotate toward the [010] direction, while those accelerating the current remain unchanged. From a statistical analysis, the dipoles hindering the electric current (with vectors of $[0\bar{1}4]/[0\bar{1}\bar{4}]$) undergo an average rotation of 10° (from ~15° at 0 V to ~5° at 0.4 V), while those accelerating the electric current (with vectors of $[014]/[01\bar{4}]$) experience almost no rotation, as shown in Fig. 3i. This unexpected observation once again rules out a pure thermal effect of the electric current, where a temperature increase typically leads to random dipole orientation, irrelevant to the direction of the current.

DFT simulations of the $\alpha$-Ag$_2$Se structure under an applied external electric field suggest a comparable alternating dipole pattern, reinforcing the STEM observations (Fig. S14). This further confirms the athermal effect of the electric current. The results clearly highlight a phenomenon akin to piezoelectric effect, where the reorientation of the dipoles induces elastic deformation within the Se-lattice.

**Phase transition induced by the electric current**
We directly observed the phase transition from the antipolar $\alpha$-Ag$_2$Se to the non-polar $\beta$-Ag$_2$Se phase, triggered by the application of a moderate voltage of 0.5 V with an electric current density of $2.15 \times 10^9$ A/m$^2$ (Step II) (Fig. S15 and Supplementary Movies 9). This observation provides valuable insight into the atomic level mechanism behind the electromechanical strain at an elevated current. The $\beta$-Ag$_2$Se phase adopts a *fcc* lattice with space group $Fm\bar{3}m$ ($a = 5.3357$ Å)[34]. The phase transition can be identified through the SAED patterns collected at different applied voltages, as shown in Fig. S16. A high-resolution image of the interface between the [100] $\alpha$-Ag$_2$Se and the [111] $\beta$-Ag$_2$Se is highlighted in Fig. S17. By analyzing the polar vector map of $\alpha$-Ag$_2$Se (Fig. S18), we measured the magnitude and angle of the dipoles in each layer adjacent to the interface (Fig. S19).

Notably, the magnitude of the dipoles at the interface clearly increases, coupled with a similar tendency for the ionic displacements within $\alpha$-Ag$_2$Se. We measured the spacing of the (001) Se-layer (as labeled in Fig. 4a) and identified an abnormal change in the first layer, ~180 and ~220 pm, compared to other layers (~150 and ~250 pm). The abnormal displacive behavior of Ag$^+$-ions at the interface suggests that these ions have gained sufficient kinetic energy in response to the applied voltage, thus enabling them to migrate toward adjacent locations where Ag$^+$-ion vacancies exist (Fig. S20). DFT calculations verify that the migration barrier of Ag-ion is ~37 times smaller than that of Se (Fig. S21).

Under the electric current induced by a moderate voltage in the ion-conductive semiconductor Ag$_2$Se, the Ag$^+$ ions migrate to neighboring Ag$^+$ vacancies (as depicted in Fig. S20), leading to the collapse of the spontaneous polarization in $\alpha$-Ag$_2$Se and triggering a phase transition to $\beta$-Ag$_2$Se. Here, we believe that the temperature rise due to the Joule heating also prompts the migration of the Ag$^+$ ions (as shown in Fig. S4). It should be noted that exposing the sample to a voltage exceeding the threshold for long-distance migration of Ag$^+$-ions will lead to severe loss of Ag, causing an irreversible phase transition. This would also make the current-induced expansion and contraction irreversible[35]. Therefore, utilizing an alternating voltage and maintaining it below the threshold voltage ($V_{transport}$) that triggers long-distance transport is essential to ensure reversible expansion and contraction, i.e., reversible $\alpha$-$\beta$ phase transitions.

The high symmetry, cubic $\beta$-Ag$_2$Se is non-polar, in which the Ag$^+$ ions occupy any of the three equivalent Ag sites. The electrically induced phase transition (from $\alpha$-Ag$_2$Se to $\beta$-Ag$_2$Se) can be characterized as a transition from an antipolar to a non-polar phase. This configuration can be visualized as the Ag$^+$ ions situated at the center of the Se-tetrahedron, as shown in the inset of Fig. 4b. Following the collapse of the dipoles, the

bonding force of the short Se-Se bonds weakens, leading to an expansion of the Se sublattice. Indeed, during the $\alpha$-to-$\beta$ transition, the projected Se-Se bond length elongates from ~3.97 Å to ~4.15 Å (as labeled in Fig. 4b), corresponding to a ~5% strain along the [001] direction, very close to the theoretically calculated strain value (~5.7%, as shown in Fig. 4e). The local strain is smaller than the experimentally observed electromechanical strain, due to the presence of defects such as interfaces of $\alpha$ and $\beta$ phases as discussed in the following section.

From the perspective of the atomic structure, once the electric current surpasses the threshold electric current density $J_T$, a migration of $Ag^+$-ions from ordered to disordered sites occurs, resulting in a large expansion of the Se-sublattice and a giant electromechanical strain. Assuming $\alpha$-Ag$_2$Se at 0 V is strain-free, we employed the GPA method on the STEM images to calculate the strain, as shown in Fig. 4c (strain along the [011] direction, which can be defined as a shear strain) and Fig. 4d (strain along the [001] direction). These images clearly reveal the generation of a substantial strain concurrent with the phase transition. Upon reducing the applied current, a reversible phase transition ($\beta$-to-$\alpha$) occurs.

### An adaptive mechanism for accommodating the elastic deformation and achieving efficient conversion of electrical energy to mechanical energy

The phase transition from $\alpha$ to $\beta$ with increasing voltage occurs gradually. Initially, nano-sized slabs of the $\beta$ phase emerge within the $\alpha$-Ag$_2$Se, followed by the transformation of the majority of the material into the $\beta$ phase, while only a small fraction of the $\alpha$ phase remains in the form of slabs (see Fig. S22). The $\alpha$-Ag$_2$Se slabs embedded in the $\beta$-phase matrix become progressively thinner as the applied voltage is further increased (Fig. S22b). Ultimately, at a voltage of 0.435 V, the entire sample completely transforms into the $\beta$-phase (Fig. S22c). Similarly, during the $\beta$ to $\alpha$ transition (with decreasing voltage), slabs of the $\beta$-phase are observed in the $\alpha$-phase matrix (Fig. S23). This heterogeneous structure, characterized by the intergrowth of $\alpha$ and $\beta$ phases, provides a flexible mechanism to accommodate large deformations during the phase transition. Meanwhile, the exact fraction of secondary phase slabs to the main phase matrix is variable, contributing to the slight difference in strain from one cycle to another.

The antipolar $\alpha$-Ag$_2$Se possesses an exceptional electromechanical strain characteristic, exhibiting both spontaneous polarizations and a low migration barrier for silver ions. In contrast to conventional ferroelectric materials, the cationic lattice within $\alpha$-Ag$_2$Se presents a low energy barrier, facilitating the displacement, hopping, and even migration of $Ag^+$ among vacancies, while the anionic lattice of $Se^{2-}$ remains relatively stable. Consequently, The electric current can induce a substantial displacement of cations, leading to noticeable changes in polarization and strain. This unique interplay between ion displacement and polarization/strain enables the polarized semiconductor Ag$_2$Se to effectively convert electrical energy into mechanical energy. Considering the potential of Ag$_2$Se in the context of flexible semiconductors for wearable and adaptable thermoelectric devices, the discovery of its giant electromechanical strain renders it applicable to a wide range of flexible electroactive devices.

## Discussion

In summary, we discovered a substantial alteration of spontaneous dipoles and a phase transition induced by electric current within the Ag$_2$Se semiconductor. The underlying atomic mechanism was comprehensively investigated using in-situ STEM. In the case of electromechanical strain caused by electric current at a lower voltage range, the slight displacement of Ag+ ions leads to changes in polarization states (Step I). As the voltage continues to increase, $Ag^+$ ions migrate to adjacent vacancies, causing dipole instability, which in turn triggers a phase transition from the antipolar orthorhombic phase $\alpha$-Ag$_2$Se to the non-polar cubic phase $\beta$-Ag$_2$Se (Step II). This transition is accompanied

by a significant expansion of the Se sublattice (Step II). Both Step I and Step II are influenced by a combination of thermal and athermal effects of the current. The findings of this research pave the way for emerging opportunities for low bandgap semiconductors, involving the integration of mechanical actuation and sensing microstructures into semiconductor platforms while preserving their flexibility and electric conductivity. These advancements extend beyond the conventional piezoelectric effect observed in ferroelectric materials.

## Methods

### Materials

Ag$_2$Se single crystals were prepared by a Se-vapor transfer method. First, Ag$_2$Se polycrystalline powders were obtained through a room-temperature dissociation adsorption reaction. After simple compaction, a sheet of Ag$_2$Se polycrystalline powders and a Se lump were vacuum sealed in a long quartz ampoule that was placed in a tube furnace with a temperature of 773 K for 5 days. Driven by the temperature difference between the outer wall and the center of the quartz ampoule, the Ag ions in Ag$_2$Se migrate to the surface where they react with the Se vapor to form large-size Ag$_2$Se crystals.

### Instrumentation

The bulk and stoichiometric Ag$_2$Se were cut and thinned down by a FIB using (Helios Nanolab G3 UC, FEI), and then transferred onto a chip. After thinning and cleaning, the sample was investigated by TEM (Talos F200s, FEI) and double C$_S$-corrected scanning/TEM (Titan Themis G2 60-300, FEI). The chip with the thin Ag$_2$Se film was loaded onto the in-situ heating and electrical holder (DENSsolutions) for in-situ experiments. The impedance spectroscopy test was carried out using a high-temperature dielectric impedance-temperature spectrometer (DMS1000), while the PFM test was conducted with a Bruker atomic force microscope (Dimension ICON-IR).

### DFT calculation

Ab initio calculations were performed using the VASP (Vienna Abinitio Simulation Package). The Perdew–Burke–Ernzerhof generalized gradient approximation was selected for the electronic exchange–correlation function. The energy cutoff for the plane-wave basis expansion was chosen as 600 eV. An energy difference of $1.0 \times 10^{-6}$ eV/atom was set to obtain accurate electronic ground-state calculation. The maximum force tolerance was set to 0.01 eV/Å for structural optimization. The Brillouin zone was sampled using the Monkhorst-Pack grid of $10 \times 10 \times 10$.

## Data availability

All data are available in the main Article and Supplementary Information, or from the corresponding author upon a reasonable request. Source data are provided with this paper.

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

## Acknowledgements

This work was supported by the National Natural Science Foundation of China (52150710537). J.W.

## Author contributions

H.L. and Q.L. were responsible for the synthesis and characterization of the materials, collection of experimental data, and revisions of the manuscript. J.W. and S.Z. conceived the project, planned the experimental procedures, analyzed the data, and wrote the manuscript. G.T. and X.W. reviewed and revised the manuscript. Y.H. and S.L. conducted the DFT simulations. Y.Z. and D.Y. participated in the synthesis of the materials and contributed to the revision of the manuscript. X.T., C.U., and Q.Z. jointly conceived the project and made improvements to the manuscript. A.G., Y.Y., and H.P. conducted an in-depth analysis of the experimental data and participated in writing the manuscript. X.G. and X.S. played a significant role in the conception of this project and provided crucial final revisions to the manuscript. All authors engaged in thorough discussion and exchange before the manuscript was submitted.

## Competing interests

The authors declare no competing interests.
