## [Transparent Peer Review file · Nature Communications]

Current induced electromechanical strain in thin antipolar Ag₂Se semiconductor

Corresponding Author: Professor Jinsong Wu

Version 0:

Reviewer comments:

Reviewer #1

(Remarks to the Author)

General comment

The manuscript is a revisited version of a previously rejected paper, which reports evidence of electromechanical effects in Ag₂Se single crystal via electrical stimuli. The analysis mainly verges on in-situ TEM analysis and proves that the electromechanical coupling occurs via phase transitions between the alpha (orthorhombic) and beta phase (cubic). The two phases have different densities of the crystals (8.3 g/cm³ vs 7.7 gr/cm³). The phase transformation is thus expected to bring a chemical expansion and contraction of approximately 6-7%, the value also reported in the paper as the maximum strain. The manuscript collects several results, and the supplementary materials describe many details in the experimental procedures and calculations. The work is undoubtedly exciting and should be published publication. However, the main text still poses some ambiguities, and some of the statements are still not fully supported by the results. The explanation of the physical mechanism especially remains confusing. These points must be addressed to accept the publication on Nature Comm.

The authors still mention antiferroelectric- and piezoelectric-like effects, implying that the material is a ferroelectric semiconductor. While ferroelectric semiconductivity is possible in wide-band gap semiconductors, Ag₂Se is less likely. The main problem is the symmetry of the crystal. While spontaneous polarisation is impossible for the cubic phase as it is centrosymmetric, the orthorhombic phase of Ag₂Se has a lower symmetry and lacks a centre of symmetry. However, this does not automatically imply antiferroelectric or piezoelectric behaviour. Effects of residual polarisation must be reported to adopt ferroelectric formalism. The term antipolar typically applies to specific structures with a regular arrangement of antiparallel dipoles, which could reconfigure, e.g., via rotational effects under the electric field. However, as the material is either an ionic conductor or a semiconductor, such effects cannot occur. This ambiguity has to be resolved between the set of properties of the phases and the electromechanical mechanism.

To be more specific, in the conclusion, the authors write:

"The electric current can induce a substantial displacement of cations, leading to noticeable changes in polarization and strain. This unique interplay between polarization and ion-conductivity enables the polarized and ion-conducting semiconductors to effectively convert electrical energy into mechanical energy....

The electromechanical strain induced by the electric current exhibits similarities to piezoelectric strain, showing both thermal and athermal effects (Step I). With a further increase of the voltage, Ag⁺-ions migrate into neighboring Ag⁺-vacancies, destabilizing the dipoles and triggering a phase transition from antipolar α -Ag₂Se to nonpolar β -Ag₂Se. This transition is accompanied by a significant expansion of the Se sublattice (Step II).".

These sentences are somehow contradicting.

#1. The authors imply that the material is a semiconductor and an ionic conductor simultaneously.

Cation displacement under the electric field does not necessarily imply that there is an ionic transport. Moreover, according to the described mechanism, the expansion should be irreversible in the superionic cubic configuration, where silver transport (migration under an electric field) can occur. In this case, the electrodes will play a major role. This possible effect is not discussed in the paper.

#2. The authors also imply that ionic conductivity occurs across the phase transition and do not explain the role of n-type

conductivity in the mechanism.

The most reasonable explanation is that a topotactic transition is induced by cation displacement to vacancy sites, probably driven by an electronic reconfiguration of the shell under the electric field. In any case, to clarify, the author should include the a macroscopic characterization of the complex dielectric properties of the material. The interplay between electronic conductivity, ionic conductivity (?) and polarization is generally characterized via impedance spectroscopy. This technique also allows the screening of frequency's effect on the dominant mechanisms with different time constants. Despite the new measurements reported in this new version, the authors do not discuss how the different frequencies activate the electromechanical coupling mechanism.

#3. On the contrary to what is stated in the paper, the piezoelectric strain does not have thermal and athermal effects. Pure piezoelectricity relies on non-centrosymmetric features of the crystals that are absent in the cubic form. Thermal dissipation is generally analyzed separately as losses in the dielectric. Moreover, piezoelectricity is a macroscopic effect that should be observable in the whole sample. The quantification of the electromechanical coupling of 6.7% is still related to lamellas samples, which is probably still a local effect due to filament formation that the authors did not exclude. Laser Doppler vibrometry or at least PFM measurements in various regions seem necessary to support the claims. I suggest the authors reflect that if the sample cannot be characterized macroscopically, it cannot be used in a real-life application.

Other Comments

- The authors write: "Electromechanical coupling is mostly observed as a piezoelectric effect in low symmetric dielectric materials". This sentence is not correct. Piezoelectricity arises in Non –centrosymmetric structures only.
- The authors write: "At temperature of ~400 K, the orthorhombic α -Ag₂Se transforms into the cubic β -Ag₂Se phase, exhibiting ion-conductivity"; The authors probably refer to the superionic Ag-conductivity. Please specify which kind of ionic conductivity. Moreover, while Ag₂Se does undergo a phase transition from the orthorhombic to a cubic structure, this transition is related to ionic conductivity and not to any form of dipole ordering typical of antiferroelectric materials. The cubic Ag₂Se has no alternating dipole structure supporting antiferroelectricity. If the authors refer to the orthorhombic phase, it should be mentioned which specific space group, in principle, can have antiferro features. Moreover, the literature also reports monoclinic phases. This additional configuration could arise under mechanical strain.
- Current reorganization of the dipoles. Please explain. What is the charge carrier? I understand there is Ag ions displacement and n-type conductivity. Therefore, the current should be the electronic current. The analysis does not support the conclusion that the electronic current induces a dipole reorganization.
- The sentence in the brackets: "(in the subsequent sections, we will prioritize discussing voltage over current density, because we directly adjusted the applied voltage during measurement, even though the strain mechanism is rooted in current density)" should be removed or reformulated to be meaningful for the reader.

Version 1:

Reviewer comments:

Reviewer #1

(Remarks to the Author)

I appreciate the care you took in revising the paper. However, while the authors have exhaustively addressed my questions and comments, they do not match them with a due revision of the main text.

The corrections in the main text are minimal and do not reflect the extensive revision claimed in the rebuttal letter. The additional experiments on macroscopic and microscopic properties to support their claims, including PFM, Impedance, and image analysis of the macroscopic features of the samples, have not been included in the discussion or the SI. I, therefore, recommend that the authors include Figure R1-R4 in the supplementary material, with the due discussion in the main text. Additional comments are:

1. In the abstract and the discussion, the authors must mention that the macroscopic strain is lower than 6.7%. What happens locally, with the TEM analysis, is not necessarily the actual property of the material.
2. There are typos ("A." 3rd line) and hyperbolic language in the abstract ("a huge electroelastic deformation"). Please rephrase.
3. The main manuscript often presents a confusing use of the past, present and future tenses. Please revise the English once more.

Version 2:

Reviewer comments:

Reviewer #1

(Remarks to the Author)

The paper is now suitable for publication.

Responses to Reviewers' comments

Review: Giant electromechanical strain triggered by current in antipolar Ag₂Se semiconductor. (NCOMMS-24-47222A)

By H. Luo, Q. Liang, *et al.*

Dear Editor and Reviewers:

We express our sincere gratitude to the reviewers for dedicating their time and effort to carefully review our manuscript and provide insightful comments. We truly appreciate their valuable suggestions. We have diligently revised our manuscript in response to their feedback, and the point-by-point responses to their comments are enclosed.

Referee #1 (Remarks to the Author):

The manuscript is a revisited version of a previously rejected paper, which reports evidence of electromechanical effects in Ag₂Se single crystal via electrical stimuli. The analysis mainly verges on in-situ TEM analysis and proves that the electromechanical coupling occurs via phase transitions between the alpha (orthorhombic) and beta phase (cubic). The two phases have different densities of the crystals (8.3 g/cm³ vs 7.7 gr/cm³). The phase transformation is thus expected to bring a chemical expansion and contraction of approximately 6-7%, the value also reported in the paper as the maximum strain. The manuscript collects several results, and the supplementary materials describe many details in the experimental procedures and calculations. The work is undoubtedly exciting and should be published publication. However, the main text still poses some ambiguities, and some of the statements are still not fully supported by the results. The explanation of the physical mechanism especially remains confusing. These points must be addressed to accept the publication on Nature Comm.

Author Reply: We greatly appreciate your positive comments and valuable insights. We also clarify the ambiguities and corrected the confused and unsupported statements in the revised manuscript, as described in detail.

1. The authors still mention antiferroelectric- and piezoelectric-like effects, implying that the material is a ferroelectric semiconductor. While ferroelectric semiconductivity is possible in wide-band gap semiconductors, Ag₂Se is less likely. The main problem is the symmetry of the crystal. While spontaneous polarisation is impossible for the cubic phase as it is centrosymmetric, the orthorhombic phase of Ag₂Se has a lower symmetry and lacks a centre of symmetry. However, this does not automatically imply antiferroelectric or piezoelectric behaviour. Effects of residual polarisation must be reported to adopt ferroelectric formalism. The term antipolar typically applies to specific structures with a regular arrangement of antiparallel dipoles, which could

reconfigure, e.g., via rotational effects under the electric field. However, as the material is either an ionic conductor or a semiconductor, such effects cannot occur. This ambiguity has to be resolved between the set of properties of the phases and the electromechanical mechanism.

Author Reply: Many thanks for your valuable comments. We agree with you that Ag_2Se is not an antiferroelectric semiconductor. The cubic phase Ag_2Se (which is thermodynamically stable when it is above 407 K) is not a ferroelectric phase as it is centrosymmetric (and there is no spontaneous polarization in the structure). The orthorhombic phase Ag_2Se exhibits anti-paralleled spontaneous polarization at room temperature. However, as the polarization cannot be reversed by an electric field or current (*i.e.* the transition from antiferroelectric-to-ferroelectric state cannot be triggered by an applied electric field). Therefore, we referred to it as “an antiferroelectric-like structure” in one case in the previous manuscript.

To avoid confusion and in response to your suggestions, we have removed all instances of ‘antiferroelectric’ in the revised manuscript when referring to Ag_2Se . Additionally, we have replaced all ‘antipolar’ with ‘a regular arrangement of antiparallel dipoles’ or simply ‘antiparallel dipoles’.

2. To be more specific, in the conclusion, the authors write:

"The electric current can induce a substantial displacement of cations, leading to noticeable changes in polarization and strain. This unique interplay between polarization and ion-conductivity enables the polarized and ion-conducting semiconductors to effectively convert electrical energy into mechanical energy....

The electromechanical strain induced by the electric current exhibits similarities to piezoelectric strain, showing both thermal and athermal effects (Step I). With a further increase of the voltage, Ag^+ -ions migrate into neighboring Ag^+ -vacancies, destabilizing the dipoles and triggering a phase transition from antipolar α - Ag_2Se to nonpolar β - Ag_2Se . This transition is accompanied by a significant expansion of the Se sublattice (Step II)."

These sentences are somehow contradicting.

Author Reply: Thanks for the valuable comments. We have revised the descriptions:

"The electric current can induce a substantial displacement of cations, leading to noticeable changes in polarization and strain. This unique interplay between ion-

displacement and polarization/strain enables the polarized semiconductor Ag_2Se to effectively convert electrical energy into mechanical energy".

"In the case of electromechanical strain caused by electric current, at lower voltage ranges, the slight displacement of Ag^+ ions leads to changes in polarization states (Step I). As the voltage continues to increase, Ag^+ ions migrate to adjacent vacancies, causing dipole instability, which in turn triggers a phase transition from the antipolar orthorhombic phase $\alpha\text{-Ag}_2\text{Se}$ to the nonpolar cubic phase $\beta\text{-Ag}_2\text{Se}$ (Step II). This transition is accompanied by a significant expansion of the Se sublattice (Step II). Both Step I and Step II are influenced by a combination of thermal and non-thermal effects of the current. "

3. The authors imply that the material is a semiconductor and an ionic conductor simultaneously.

Cation displacement under the electric field does not necessarily imply that there is an ionic transport. Moreover, according to the described mechanism, the expansion should be irreversible in the superionic cubic configuration, where silver transport (migration under an electric field) can occur. In this case, the electrodes will play a major role. This possible effect is not discussed in the paper.

Author Reply: Thanks for the valuable comments. We agree with you that cation displacement and transport under the electric field are different. While cation displacement can be triggered under a relatively small electric field, the ionic transport requires a larger electric field applied over a longer duration, especially when Ag_2Se has transformed into the superionic cubic structure^[1]. Once the long-distance transport leads to a severe loss of Ag-ion in the materials, the resulting phase transition will become irreversible. As you have noted, incorporating active electrode such as Ag and Cu can help maintain this reversible transformation. Meanwhile, both orthorhombic and cubic Ag_2Se structures exhibit certain resilience to slight loss of Ag, allowing reversible phase transformation between a Ag-deficient orthorhombic Ag_2Se and cubic Ag_2Se ^[1], as observed in our study. For practical applications, it is crucial to utilize an alternating voltage while keeping it below the threshold voltage ($V_{\text{transport}}$) that triggers long-distance transport, yet above the critical voltage needed for phase transition, which is normally smaller than $V_{\text{transport}}$ ^[2]. Under this situation, short-range migration or hopping of the silver cations occurs rather than the long-distance transportation, thus it is a reversible process which can be confirmed in our current research.

According to this comment, we have added the corresponding discussion to the revised manuscript: "It should be noted that exposing the sample to a voltage exceeding the threshold for long-distance migration of Ag^+ -ions^[2] will lead to severe loss of Ag,

causing an irreversible phase transition and making the current induce expansion and contraction irreversible. Therefore, utilizing an alternating voltage and maintaining it below the threshold voltage ($V_{\text{transport}}$) that triggers long-distance transport is essential to ensure the reversible expansion and contraction, *i.e.*, reversible α - β phase transitions.”

References:

[1] Wu, H., et al. Advances in Ag₂Se-based thermoelectrics from materials to applications. *Energy Environ. Sci.* **16**, 1870–1906(2023).

[2] Guo A., et al. Resistive Switching in Ag₂Te Semiconductor Modulated by Ag⁺- Ion Diffusion and Phase Transition. *Advanced Electronic Materials*, **8(12)**, 2200850(2022).

4. The authors also imply that ionic conductivity occurs across the phase transition and do not explain the role of n-type conductivity in the mechanism.

The most reasonable explanation is that a topotactic transition is induced by cation displacement to vacancy sites, probably driven by an electronic reconfiguration of the shell under the electric field. In any case, to clarify, the author should include the a macroscopic characterization of the complex dielectric properties of the material. The interplay between electronic conductivity, ionic conductivity (?) and polarization is generally characterized via impedance spectroscopy. This technique also allows the screening of frequency's effect on the dominant mechanisms with different time constants. Despite the new measurements reported in this new version, the authors do not discuss how the different frequencies activate the electromechanical coupling mechanism.

Author Reply: Thanks for the valuable suggestion. We agree with you that since Ag₂Se is a n-type, narrow band gap semiconductor, its major conductivity is the electronic conductance (rather than ionic conductance). Electron is the major carrier for the conductance^[1].

We examined the changes of the carrier concentration and carrier mobility in Ag₂Se as a function of temperature, as shown in Figure R1. It was observed that at T \approx 360K, the free carrier concentration increased sharply while the carrier mobility decreased. This was attributed to the migration of Ag⁺ ions, scattering the electron migration although the carrier concentration (involving Ag⁺ ions) increases.

As you kindly suggested, we measured the impedance spectroscopy trying to understand the interplay of the electronic conductivity and ionic conductivity, as illustrated in Figure R2. In the range of 298K to \sim 373K, the Nyquist plot displays a single semicircle,

indicating that electron conduction predominantly drives the conductance of the system. However, in the temperature range between 373K and 398K, while the high frequency Nyquist plot still behaves as a semi-circle, a diffusion tail appears in the low frequency region, which is the characteristic of Warburg impedance. The Warburg impedance typically indicates charge carrier diffusion at the electrode surface, suggesting that in this temperature range, in addition to electron conduction, ion migration of the system begins to occur. Therefore, it can be inferred that below 373K, the conduction mechanism of the system is mainly electronic conductance. Above 373K, the conduction mechanism changes to a mixture of electronic and ionic conductance. Meanwhile, as the electronic resistance is very small (as Ag_2Se is a narrow bandgap semiconductor), the overall polarization change in response to electronic conductance is challenging to measure (although there indeed exists spontaneous, orderly arrangement of antiparallel dipoles in the low-temperature $\alpha\text{-Ag}_2\text{Se}$ phase).

In summary, we do also observe a notable athermal effect of the electronic current. You have suggested one possible cause for this athermal effect - ‘an electronic reconfiguration of the shell under the electric field’- which aligns well with our observations.

Figure R1. The relationship between carrier concentration and carrier mobility with temperature.

Figure R2. Nyquist plot of Ag_2Se in the temperature range of 298K to 423K. Frequency range from 10^2Hz to 10^6Hz .

[1]Wu, H., et al. Advances in Ag₂Se-based thermoelectrics from materials to applications. *Energy Environ. Sci.* **16**, 1870–1906(2023).

5. On the contrary to what is stated in the paper, the piezoelectric strain does not have thermal and athermal effects. Pure piezoelectricity relies on non-centrosymmetric features of the crystals that are absent in the cubic form. Thermal dissipation is generally analyzed separately as losses in the dielectric. Moreover, piezoelectricity is a macroscopic effect that should be observable in the whole sample. The quantification of the electromechanical coupling of 6.7% is still related to lamellas samples, which is probably still a local effect due to filament formation that the authors did not exclude. Laser Doppler vibrometry or at least PFM measurements in various regions seem necessary to support the claims. I suggest the authors reflect that if the sample cannot be characterized macroscopically, it cannot be used in a real-life application.

Author Reply: Many thanks for the valuable comments. We agree with you that piezoelectric strain does not have thermal and non-thermal effects.

In the revised manuscript, we clarify that the current has both thermal and athermal effects. The piezoelectric-like effect we identified in the electromechanical coupling arises from the non-centrosymmetric features of the orthorhombic structure. We use the term “piezoelectric-like” instead of piezoelectric, to differentiate it from typical piezoelectric effects occurred in a dielectric crystal. This phenomenon was further validated through macroscopic tests (Figure R3, about 1.8% strain), where applying a 10V voltage to the sample produced observable strain and an elastic deformation, the measured temperature, 346.4K, remained well below the theoretical phase transition temperature of 406K.

Additionally, we conducted PFM experiments following your suggestion. Fig R4 present the amplitude and phase signals, showing an obvious phase lag and amplitude butterfly loop, indicating the measured strain under applied electric field.

Figure R3. Macroscopic characterization of the elastic deformation of Ag_2Se under the applied voltage/current. (a) Experimental setup consisting of a copper clamp, a conductive copper (Cu) electrode in the middle and a thermocouple placed at the bottom. The deformation was captured using the optical microscope and a CCD camera. (b) The Ag_2Se block used in the testing has dimensions of 3 x 2 x 1.5 mm. (c) The morphology of the Ag_2Se sample before the application of the voltage. (d) The morphology of the Ag_2Se sample when the voltage of 10 V is applied, A significant strain was observed in the images. (e, f) Measured temperatures corresponding to figures (c) and (d).

Figure R4. Amplitude and phase signal of PFM.

6. The authors write: "Electromechanical coupling is mostly observed as a piezoelectric effect in low symmetric dielectric materials". This sentence is not correct. Piezoelectricity arises in Non –centrosymmetric structures only.

Author Reply: Many thanks for the good suggestion. We have rewritten the sentence as: "Electromechanical coupling is mostly manifested as a piezoelectric effect in non-centrosymmetric dielectric materials".

7. The authors write: "At temperature of ~ 400 K, the orthorhombic α -Ag₂Se transforms into the cubic β -Ag₂Se phase, exhibiting ion-conductivity"; The authors probably refer to the superionic Ag-conductivity. Please specify which kind of ionic conductivity. Moreover, while Ag₂Se does undergo a phase transition from the orthorhombic to a cubic structure, this transition is related to ionic conductivity and not to any form of dipole ordering typical of antiferroelectric materials. The cubic Ag₂Se has no alternating dipole structure supporting antiferroelectricity. If the authors refer to the orthorhombic phase, it should be mentioned which specific space group, in principle, can have antiferro features. Moreover, the literature also reports monoclinic phases. This additional configuration could arise under mechanical strain.

Author Reply: Many thanks for the valuable suggestions. We intended to refer specifically to 'the superionic Ag-conductivity'. In the revised manuscript, we have corrected the sentence to 'At temperature of ~ 400 K, the orthorhombic α -Ag₂Se transforms into the cubic β -Ag₂Se phase, exhibiting superionic Ag-conductivity'.

We agree with your observation that cubic Ag₂Se is not antiferroelectric and lacks a dipole structure. However, we did observe regularly arranged antiparallel dipoles in the α -Ag₂Se phase, which we described as a structure with antiparallel dipoles (or an 'antiferroelectric-like phase') rather than a true antiferroelectric phase. The regularly arranged antiparallel dipoles in the α -Ag₂Se phase disappear upon transition to the cubic β -Ag₂Se phase, a phase transition analogous to an antiferroelectric-to-paraelectric phase transition. Notably, the monoclinic phase structure is absent in our sample (Our sample is a large orthorhombic phase single crystal with size of $4\mu\text{m}\times 4\mu\text{m}\times 0.15\mu\text{m}$).

8. Current reorganization of the dipoles. Please explain. What is the charge carrier? I understand there is Ag ions displacement and n-type conductivity. Therefore, the current should be the electronic current. The analysis does not support the conclusion that the electronic current induces a dipole reorganization.

Author Reply: Thanks for the valuable question. As n-type semiconductor with a narrow bandgap, the primary charge carrier is the electron (with a negative charge). The electronic current here refers to the energy carried by the electron. In the superionic phase structure, conduction involves both ionic and electronic contributions, with

electronic conductance being dominant [1]. The experiments (especially those performed at low temperatures) indicate that there is dipole reorganization (due to the Ag ion displacement) under an electric current (when voltage is applied). We did experimentally observe that the dipoles' magnitudes (extending along the current direction) and their orientation (aligned with the vortex pattern formed by the dipoles) change as current flows over the sample. The polarization vector diagram of α -Ag₂Se at 0 V and 0.4 V was compared. Polarization along the electric field: in the current direction, the length of the polarization vector $[0\bar{1}4]/[014]$ increases from ~ 54 pm to ~ 58 pm, indicating increased polarization. Polarization in the direction of the inverse electric field: in the direction of the inverse current, the length of the polarization vector $[01\bar{4}]$ and $/ [0\bar{1}\bar{4}]$ decreases from ~ 68 pm to ~ 63 pm, indicating a weakening of polarization. A statistical analysis shows the magnitude and angle of the dipoles measured at 0 V and 0.4 V. The dipoles that hinder the current rotate an average angle of 10° (from $\sim -15^\circ$ at 0 V to $\sim -5^\circ$ at 0.4 V), while those that accelerate the current do not rotate (Figure R5). As you have kindly pointed out, it is likely that the athermal effect of the current is arisen due to an electronic reconfiguration of the shell under the electric field.

Figure R5 (a) STEM image overlapping with the dipoles oriented along the electric field (those with vectors of $[0\bar{1}4]$ and $[014]$), at 0 V (blue) and 0.4 V (purple). The dipoles are elongated with an average magnitude changing from ~ 54 pm to ~ 58 pm. (b) STEM image overlapping with the dipoles oriented against the electric field (those with vectors of $[01\bar{4}]$ and $/ [0\bar{1}\bar{4}]$), at 0 V (blue) and 0.4 V (purple). The dipoles are shortened (the average magnitude changes from ~ 68 pm to ~ 63 pm). (c) A statistical analysis showing the magnitude and angle of the dipoles measured at 0 V and 0.4 V. The dipoles that hinder the current rotate an average angle of 10° (from $\sim -15^\circ$ at 0 V to $\sim -5^\circ$ at 0.4 V), while those that accelerate the current do not rotate. All scale bars are 1 nm.

[1]Wu, H., et al. Advances in Ag₂Se-based thermoelectrics from materials to applications. *Energy Environ. Sci.* **16**, 1870–1906(2023).

9.The sentence in the brackets: "(in the subsequent sections, we will prioritize discussing voltage over current density, because we directly adjusted the applied voltage during measurement, even though the strain mechanism is rooted in current density)" should be removed or reformulated to be meaningful for the reader.

Author Reply: Thanks for the good suggestion and sorry for the confusion. We have revised the sentence as "we will use voltage as the driving parameter in the subsequent sections, as we directly adjusted the applied voltage, rather than the current density, in the experiment".

Responses to Reviewers' comments

Review: Giant electromechanical strain triggered by current in antipolar Ag₂Se semiconductor. (NCOMMS-24-47222A)

By H. Luo, Q. Liang, et al.

Dear Editor and Reviewers:

We express our sincere gratitude to the reviewers for dedicating their time and effort to carefully review our manuscript and provide insightful comments. We truly appreciate their valuable suggestions. We have diligently revised our manuscript in response to their feedback, and the point-by-point responses to their comments are enclosed.

Reviewer #1 (Remarks to the Author):

1. The corrections in the main text are minimal and do not reflect the extensive revision claimed in the rebuttal letter. The additional experiments on macroscopic and microscopic properties to support their claims, including PFM, Impedance, and image analysis of the macroscopic features of the samples, have not been included in the discussion or the SI. I, therefore, recommend that the authors include Figure R1-R4 in the supplementary material, with the due discussion in the main text.

Author Reply: Really appreciate the valuable comments. Following the reviewer's kind suggestions, we have added all of the related contents (including Figures R1-R4 and relevant discussions) to the main content or SI in the revised manuscript. Some of them are copied below: (Relevant modifications are marked in yellow in the marked version.)

For Figures R1-2 (which is Figure S5 in the revised SI): “To further explore the interaction between electronic conductivity and ionic conductivity in Ag₂Se, we tested the dielectric properties, carrier mobility and carrier concentration, as shown in Figure S5. As seen in Figure S5a, when the temperature reaches approximately 360K, the free carrier concentration of Ag₂Se increases sharply, accompanied by a decrease in carrier mobility. This phenomenon is attributed to the migration of Ag⁺ ions, which scatter electrons and thus affect electrical conductivity. From 298K to 373K, the Nyquist plot shows a single semicircle, indicating that electronic conduction dominates. Above 373K, the high-frequency region of the Nyquist plot remains semicircular, while a diffusion tail appears in the low-frequency region (Warburg impedance), suggesting the onset of ionic migration. Thus, below 373K, the main conduction mechanism is electronic, while above 373K, it becomes a hybrid of electron and ion conduction (Figure S5b)”

For Figures R3 and R4 (which are Figures S10 and S11, respectively, in the revised SI): “The strain was further verified by macroscopic tests (Figure S10), which showed a strain of about 1.8%. In these tests, a voltage of 10V was applied to the sample, producing observable strain and elastic deformation at a temperature of 346.4K, which is significantly lower than the theoretical phase transition temperature of 406K. Additionally, piezoelectric force microscopy (PFM) experiments were carried out. Figure S11 presents the amplitude and phase signal, confirming the presence of strain

under the applied electric field.”

Also, descriptions have added to Experimental: “The impedance spectroscopy test was carried out using a high-temperature dielectric impedance-temperature spectrometer (DMS1000), while the Piezoelectric force microscopy (PFM) test was conducted with a Bruker atomic force microscope (Dimension ICON-IR).”

2. In the abstract and the discussion, the authors must mention that the macroscopic strain is lower than 6.7%. What happens locally, with the TEM analysis, is not necessarily the actual property of the material.

Author Reply: Thanks for the good comments, we have added relevant content in the abstract and summary, “leading to a giant local strain of 6.7% measured by in situ transmission electron microscopy (which may not necessarily reflect the property of the bulk material).”. (Relevant modifications are marked in yellow in the marked version.)

3. There are typos (“A.” 3rd line) and hyperbolic language in the abstract (“a huge electroelastic deformation”). Please rephrase.

Author Reply: Thanks for the good suggestion. We have revised the typos and hyperbolic language in the abstract to ensure the accuracy of the data.

“A.” → it has been deleted.

The writing “a huge electroelastic deformation” has been rephrased as “leading to a giant local strain of 6.7% measured by in situ transmission electron microscopy (which may not necessarily reflect the property of the bulk material).”.

We have also corrected other typos:

“*i.e.*, ~423K in our experiments²³” → “*i.e.*, ~407K²³”

“The antipolar α -Ag₂Se possesses an exceptional characteristic in terms of electromechanical strain, exhibiting both spontaneous polarizations and high ionic mobility.” → “The antipolar α -Ag₂Se possesses an exceptional electromechanical strain characteristic, exhibiting both spontaneous polarizations and a low migration barrier for silver ions. ”

(Relevant modifications are marked in yellow in the marked version.)

4. The main manuscript often presents a confusing use of the past, present and future tenses. Please revise the English once more.

Author Reply: Thanks for the thorough review and the good comments. We have carefully checked and revised the tenses in the manuscript (which are highlighted in the marked version of the revised manuscript), and have refined the language throughout the entire paper. All corrections related to tenses and English writing are marked in green in the marked version.

Responses to Reviewers' comments

Review: Giant electromechanical strain triggered by current in antipolar Ag₂Se semiconductor. (NCOMMS-24-47222A)

By H. Luo, Q. Liang, et al.

Dear Editor:

We express our sincere gratitude to you and the reviewers for dedicating your time and effort to carefully handle and review our manuscript. We have incorporated all of the editorial requirements (as list in the *Author Checklist*). We have included the point-by-point responses to the reviewer's comments (as copied below).

Reviewer #1 (Remarks to the Author):

1. The paper is now suitable for publication.

Author Reply: We extend our heartfelt gratitude for the revisions suggested by the reviewer. Following your advice, the article and related materials have been appropriately refined to meet the publication standards. We sincerely appreciate your efforts in enhancing the quality of the paper.

.